

# Analysis of how the spatial and temporal patterns of fire and their bioclimatic and anthropogenic drivers vary across the Amazon rainforest in El Niño and non-El Niño years

Minerva Singh[1] and Xiaoxiang Zhu[2]

[1] Imperial College London, LONDON, United Kingdom
[2] Technical University of Munich, Munich, Germany

Corresponding author
Minerva Singh,
minerva_singh@yahoo.co.in

## ABSTRACT

In the past two decades, Amazon rainforest countries (Brazil, Bolivia, Colombia, Ecuador, Guyana, Peru and Venezuela) have experienced a substantial increase in fire frequency due to the changes in the patterns of different anthropogenic and climatic drivers. This study examines how both fire dynamics and bioclimatic factors varied based on the season (wet season and dry season) El Niño years across the different countries and ecosystems within the Amazon rainforest. Data from publicly available databases on forest fires (Global Fire Atlas) and bioclimatic, topographic and anthropogenic variables were employed in the analysis. Linear mixed-effect models discovered that year type (El Niño *vs.* non-El Niño), seasonality (dry *vs.* wet), land cover and forest strata (in terms of canopy cover and intactness) and their interactions varied across the Amazonian countries (and the different ecosystems) under consideration. A machine learning model, Multivariate Adaptive Regression Spline (MARS), was utilized to determine the relative importance of climatic, topographic, forest structure and human modification variables on fire dynamics across wet and dry seasons, both in El Niño and non-El Niño years. The findings of this study make clear that declining precipitation and increased temperatures have strong impact on fire dynamics (size, duration, expansion and speed) for El Niño years. El Niño years also saw greater fire sizes and speeds as compared to non-El Niño years. Dense and relatively undisturbed forests were found to have the lowest fire activity and increased human impact on a landscape was associated with exacerbated fire dynamics, especially in the El Niño years. Additionally, the presence of grass-dominated ecosystems such as grasslands also acted as a driver of fire in both El Niño and non-El Niño years. Hence, from a conservation perspective, increased interventions during the El Niño periods should be considered.

## INTRODUCTION

Fire dynamics in the Amazon basin have changed considerably over the past few decades, a consequence of changes in land cover and weather (*Schroeder et al., 2005*; *Alencar et al., 2015*; *Da Silva Júnior et al., 2019*). Fire frequency and severity in the Amazon basin are influenced by temperature, precipitation, logging and fragmentation (*Aragao et al., 2007*; *Alencar et al., 2011*; *Morton et al., 2013*; *Armenteras & Retana, 2012*) under all climate scenarios (*Abatzoglou et al., 2018*; *Gutman, Csiszar & Romanov, 2000*). Forest fires are exacerbated by droughts resulting from El Niño events (*Gutman, Csiszar & Romanov, 2000*; *Alencar, Nepstad & Diaz, 2006*; *Taufik et al., 2017*). The El Niño droughts of 2005 resulted in a 33% increase in forest fires in the south-western part of Amazonia, compared to previous years (*Aragao et al., 2007*). Below average rainfalls and higher average temperatures were observed in both eastern and central Amazonia during the El-Niño years, arguably resulting in higher fire activity (*Li et al., 2011*). The direct impacts of El Niño (*i.e.* being an El Niño year) combined with anthropogenic pressures was found to be powerful driver of elevated fire activity in the Roraima state of Brazil from 2015–2016 (*Fonseca et al., 2017*).

Global climatic patterns and localized land cover change also drive fire dynamics in the Amazon basin (*Adeney, Christensen & Pimm, 2009*). Local-scale anthropogenic disturbances degrade and fragment tropical forests, including those of the Brazilian Amazon (*Dwomoh et al., 2019*; *Andela et al., 2019*) and NW Amazonia (*Armenteras & Retana, 2012*), making them substantially more susceptible to fires. Anthropogenic land cover changes, such as deforestation-induced fragmentation induce several structural and functional changes in the Amazon basin such as the biomass collapse around the forest edges (*Junior et al., 2020*). These changes, arguably have a cascading impact on fire dynamics. Fragmentation was identified as being a major driver of high intensity in the central Brazilian Amazonia region (*Silva Junior et al., 2018*). High levels of deforestation in the Brazilian states of Acre, Amazonas and Roraima corresponded to elevated fire activity from 2003–2019 (*Silveira et al., 2020*). While the research by *Junior et al. (2020)* discovered that deforestation rates and the formation of forest edges varied across the different Amazonian countries, it still not known how these anthropogenic disturbances explain the variation in fire attributes of the different ecosystems of different countries that all form a part of the Amazonian biome. Moreover, anthropogenic disturbances interact with climatic patterns to exacerbate fire events in the Amazon. While the relative importance of climatic and anthropogenic drivers in influencing fire dynamics varies spatially (*Armenteras & Retana, 2012*; *Da Silva et al., 2018*), the nature of these interaction has not been systematically examined across the different Amazonian countries.

A multi-temporal study from southern Amazonia discovered that inter-annual fire variability was uncorrelated with deforestation and low night-time humidity was an important driver of single and recurring fires in Mato Grosso. A cross-country comparison discovered that most fires occurred close to forest edges in NW Amazonia, road construction, fragmentation and deforestation played an important role in mediating these fires (*Armenteras et al., 2017*). The majority of fire studies focus on the temporal patterns

of fire and their frequencies (*Schroeder et al., 2005*; *Morton et al., 2013*; *Armenteras & Retana, 2012*; *Da Silva et al., 2018*; *Uriarte et al., 2012*). However, fire characteristics such as size and extent and fire behavioral attributes such as speed are important components of fire regimes (*McLauchlan et al., 2020*). These characteristics can have important ramifications for forest ecosystems. The fire perimeter over multiple days generates quadratic growth in the burned area (*Page et al., 2017*). Comparatively less research has focused on the role of natural and anthropogenic factors in driving fire duration and the role seasonal variation in influencing these fire patterns. This arguably is a significant caveat in our understanding of fire dynamics, their variation across the different parts of the Amazon basin and the role of both natural and anthropogenic factors in driving the different components of fire.

The bulk of the fire dynamics related research in the Amazon basin has focused on the Brazilian Amazon and limited research has evaluated the variation in fire dynamics across the different parts of the Amazon biome (*Armenteras & Retana, 2012*; *Da Silva et al., 2018*; *Armenteras et al., 2017*). However, no systematic assessments have been taken to identify the variation in fire patterns for the different Amazonian countries. Examining the variation in fire dynamics and their natural and anthropogenic drivers and forest structure variables across the different parts of the Amazonian basin will facilitate a cross-country examination of the variation in fire dynamics and help develop country/ region-specific fire management policies.

Hence, in this study the spatial-temporal variation of relatively less studied fire characteristics, namely fire size, duration and perimeter across the different ecosystems and countries that comprise the Amazon biome (Brazil, Bolivia, Peru, Colombia, Guyana and Ecuador) were quantified. We assessed how the spatial temporal variation of these fire characteristics is influenced by bioclimatic and anthropogenic drivers and the interactions of these with the ecosystem type and structure across seasons and with El Niño/non-El Niño years. The main aims of the research are to identify: (1) If and how fire characteristics (fire size, fire speed, fire duration and expansion) and their bioclimatic drivers (precipitation, maximum temperature, soil moisture and drought severity) varied between dry and wet seasons of the Amazon biome from 2003–2016; (2) if the variations in fire and bioclimatic driver dynamics were more pronounced in the El Niño years; (3) interactions between climatic, topographic, forest structure and human modification variables and fire dynamics during the wet and dry seasons of the El Niño and non-El Niño years (from 2003–2016); and (4) the relative importance of both the anthropogenic and natural drivers in influencing the different fire attributes between the dry and wet seasons of El Niño and non-El Niño years.

## MATERIALS & METHODS

### Study area

The study region boundary is defined by the biogeographical limit of the greater Amazon ecosystem which includes ecosystems known as Amazonian biome within the seven countries—Brazil, Bolivia, Colombia, Ecuador, Guyana, Peru and Venezuela whose borders fall within this biogeographical limit (*Walker et al., 2020*), see Fig. 1.

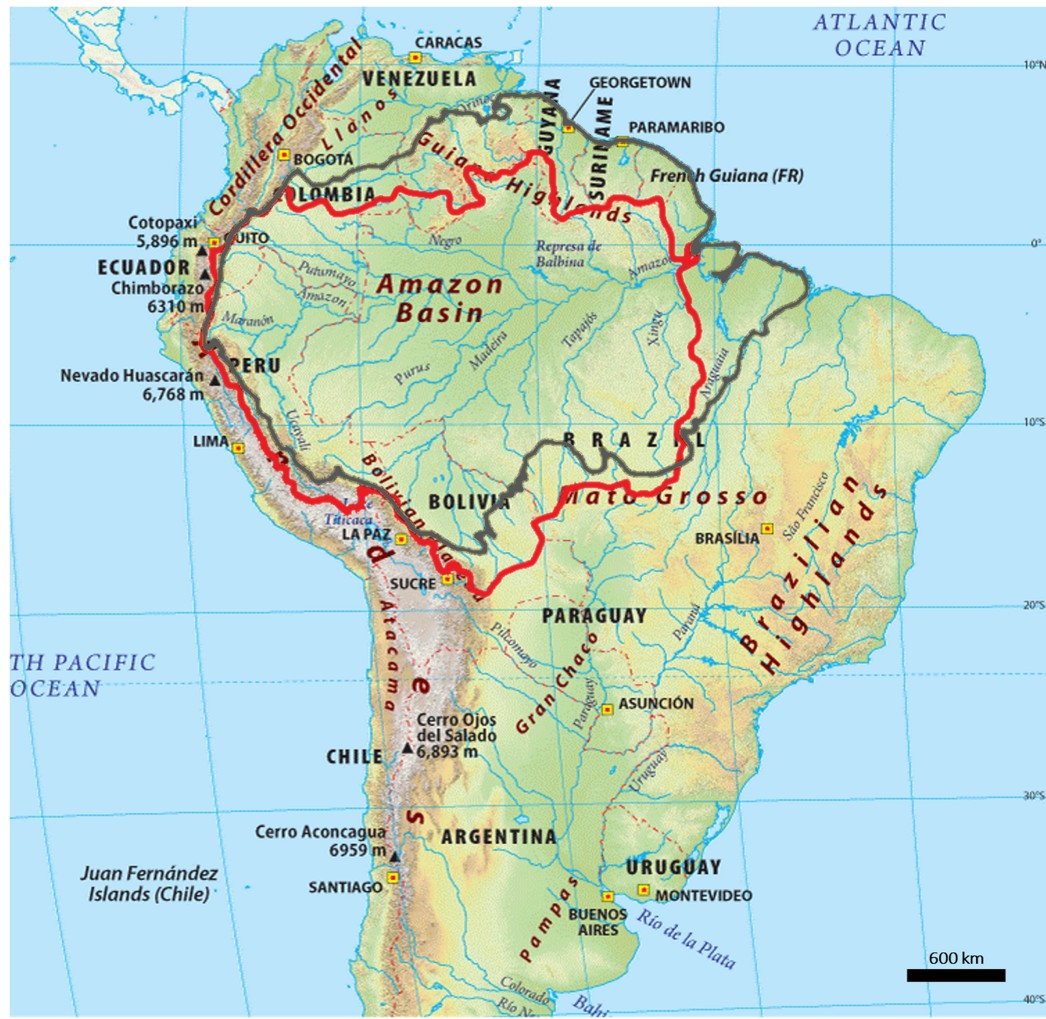

**Figure 1 Outline map of the Amazon biome (red outline) and Amazon basin (black outline).**

In terms of ecosystems, Amazon biome mostly comprises dense moist tropical forests and other smaller vegetation types including savannas, grasslands and swamps (*Devecchi et al., 2020*). Owing to a combination of agricultural and infrastructure development (*Finer et al., 2008*), many parts of the Amazon biome have undergone extensive land cover change and are now dominated by successional forests and croplands (*Nobre et al., 2016*).

## Fire data

In this research, we used the Global Fire Atlas (GFA), a global dataset of fire dynamics (from 2003–2016). GFA is a dataset that monitors the daily dynamics of individual fires across the globe. The dataset is derived from the Moderate Resolution Imaging Spectroradiometer (MODIS) daily moderate-resolution (500 m) Collection 6 MCD64A1 burned-area data. The MCD64A1 MODIS, which underpins the global fire atlas dataset, has been extensively used for fire mapping and monitoring in the tropics, including the Amazon rainforest (*Da Silva Júnior et al., 2019*; *Shimabukuro et al., 2020*; *Pereira et al.,*

**Table 1 Overview of Global Fire Atlas data layers. Shapefiles of ignition locations and fire perimeters contain attribute tables for individual fires with summary information.**

| Attribute name | Shapefile attributes | 500 m daily gridded | 0.250 monthly gridded | Description |
|---|---|---|---|---|
| Ignitions | Location and timing | – | Sum | Monthly ignition count |
| Size (km$^2$) | Per fire | – | Average | Average per fire-total area burned |
| Duration (days) | Per fire | – | Average | Average per fire duration |
| Daily fire expansion (km$^2$ day$^{-1}$) | Average per fire | – | Average | Average per fire expansion |
| Speed (km day$^{-1}$) | Fire | yes | Average | Average per fire speed |
| The direction spread (−) | Dominant per Fire | yes | Dominant | Dominant per fire direction of spread based on multi-day fires: (1) north, (2) northeast, (3) east, (4) southeast, (5) south, (6) southwest, (7) west and (8) northwest. |

2017; *Rossi & Santos, 2020*). MCD64A1 has detected most total burned areas globally (*Humber et al., 2018*). This is owing to an improved algorithm which facilitates an increase in the detection of small burns, reduces the uncertainty in the temporal detection of the day of burning, and facilitates increased detection in high-cloud areas (*Giglio et al., 2016*). GFA used two filters to account for uncertainties in the day of burn, to map the location and timing of fire ignitions and the extent and duration of individual fires. Based on the GFA algorithm, the burned area was broken down into seven fire characteristics in three steps. First, the burned area was described as the product of ignitions and individual fire sizes. Second, fire size was further separated into fire duration and a daily expansion component. Third, the daily fire expansion was subdivided into fire speed, the length of the fire line and the direction of spread (*Andela et al., 2019*).

All the MODIS grid cells at 500-m resolution were compared with corresponding active-fire detection from 2012–2016 for four different ecosystems: (1) forests (including all forests), (2) shrublands (including open and closed shrublands), (3) woody savannas, and (4) savannas and grasslands. The fire perimeters from GFA were also compared to the fire perimeter estimates from the Monitoring Trends in Burn Severity (MTBS) project from 2003–2015. The two thresholds were used to select a subset of MTBS and GFA perimeters to assess the accuracy of estimated fire duration. Fires were first matched based on perimeters, with a maximum tolerance of a threefold difference in length between perimeters. MTBS perimeters was selected with VIIRS active-fire detections for at least 25% of the 500 m Fire Atlas grid cells. Over the 14-year study period, 13,250,145 individual fires were identified with an average size of 4.4 km$^2$ (Table 1) at MODIS pixel of 21 ha or 0.21 km$^2$. While the accuracy varies with vegetation types, the product gave robust estimates for forested ecosystems (*Andela et al., 2019*). For each individual fire, data is provided on the timing and location of ignitions, fire size, perimeter, duration, daily expansion, daily fire line, speed and direction of spread. See Table 1 for details. Data available as vector shapefiles that delineate individual fire perimeters (polygon) and point of ignition (point) (Table 2) were used in this study. Dominant land cover in the data was derived from the MODIS MCD12Q1 collection 5.1 and followed the University of Maryland (UMD) classification (*Andela et al., 2019*).

**Table 2 Overview of Global Fire Atlas data layers. Shapefiles of ignition locations and fire perimeters contain attribute tables for individual fires with summary information.**

| Name | Units | Min | Max | Scale | Description |
|------|-------|-----|-----|-------|-------------|
| pdsi | | −4,317* | 3,418* | 0.01 | Palmer drought severity index |
| Pr | mm | 0* | 7,245* | | Precipitation accumulation |
| Soil | mm | 0* | 8,882* | 0.1 | Soil moisture derived using a one-dimensional soil water balance model |
| tmmn | °C | −770* | 387* | 0.1 | Minimum temperature |
| tmmx | °C | −670* | 576* | 0.1 | Maximum temperature |

**Note:**
  * Estimated min or max value.

**Table 3 El-Nino years from 2003–2016.**

| Phenomenon | Years of occurence | Intensity |
|------------|--------------------|-----------|
| El Nino | 2002–2003 | Moderate |
| El Nino | 2004–2005 | Weak |
| El Nino | 2009–2010 | Moderate |
| El Nino | 2015–2016 | Strong |

## Bioclimatic variables

TerraClimate is a monthly global gridded dataset of temperature, precipitation, and other water balance variables from 1958-present. Its combination of the high spatial resolution of ~4 km$^2$, global extent and long length fills a unique gap in climate data. Using climatically aided interpolation, TerraClimate combines spatial climatology from the WorldClim dataset, with time-varying information from the coarser-resolution CRU Ts4.0 and Japanese 55-year Reanalysis (JRA55), to produce a monthly dataset of precipitation, maximum and minimum temperature, wind speed, vapour pressure and solar radiation (Table 1).

In addition, TerraClimate uses a water balance model that incorporates temperature, precipitation, reference evapotranspiration, and interpolated plant extractable soil water capacity to produce a monthly surface water balance dataset. The procedure applies interpolated time-varying anomalies from CRU Ts4.0/JRA55 to the high spatial resolution climatology of WorldClim, creating a high spatial resolution dataset covering a broad temporal range. Temporal information on global land surface temperature, precipitation, and vapour pressure is acquired from CRU Ts4.0, whilst JRA55 data is used for regions where CRU data is lacking. See Table 2 for the details of the bioclimatic variables used in the study.

Since the research seeks to examine the variation of bioclimatic and fire dynamics for both El Niño and non-El Niño years, the following years considered as El Niño were included in the study (*Moura et al., 2019*) (Table 3):

The El Niño year categorization was taken from the work of *Moura et al. (2019)*. The period of August–October is a widely used categorization for dry season (*Yokelson et al., 2007*; *Heyer et al., 2018*; *Xu et al., 2020*; *Pope et al., 2020*) and is used in this study

since this is also the period in which a lot of burning, including deforestation burning takes place (*Moura et al., 2019*; *Morgan et al., 2019*; *Reddington et al., 2015*). The global human modification index (GHM), which exists on a scale of 0–100 was used as a proxy for anthropogenic disturbances. GHM is the cumulative measure of human modification of terrestrial lands. It utilizes the physical extents of 1394 anthropogenic stressors and their estimated impacts using spatially explicit global datasets with the median year of 2016 and spatial resolution of 1-km$^2$. Higher GHM values indicate greater human modification (*Kennedy et al., 2019*). In addition to land cover and human modification, the study also included forest structure or strata information obtained from (*Tyukavina et al., 2015*). Forest strata were defined in terms of forest structural characteristics and comprised the year 2000 percent tree canopy cover obtained from (*Da Silva Júnior et al., 2019*), tree height (*Tyukavina et al., 2015*) and forest intactness (*Potapov et al., 2008*). The forest strata were categorized as—(1) Strata 1: low canopy cover (2) Strata 2: medium canopy cover-short height (3) Strata 3: medium canopy cover-tall height (4) Strata 4: dense canopy cover-short height (5) Strata 5: dense canopy cover-short height-intact structure (6) Strata 6: dense cover-tall height (7) Strata 7: dense cover-tall height- intact structure (8) Strata 8: barren earth (*Tyukavina et al., 2015*).

## Data analysis

To evaluate the impact of El Niño years, seasonality, landcover, forest strata and the interaction between these on the bioclimatic and fire dynamic variables, general linear mixed effect model was implemented using the R package nlme (*Jones & DuVal, 2019*) for each of the countries. These models can determine the general effect of a process on a group of individuals, as well as the degree to which each individual differs from the general response (*Archibald et al., 2010*). The El Niño and non-El Niño years were incorporated into the model by assigning each unique incidence an identifier, which corresponded to a temporal block which ensures each of the El Niño/Non-El Niño events are treated as replicates for the tests of El Niño effects. This temporal block was taken as a random effect and this accounts for similarity of within-block samples closer together in time, analogous to spatial blocking to account for autocorrelation.
The response variables were linearized *via* data transformations and diagnostic plots of residuals *vs.* predicted values indicated that these transformations effectively decoupled the variance and mean, so the normal errors model lme was chosen. Post-hoc analysis was used using the emmeans package of the R programming language (*Team, 2013*).
This package allows the implementation of marginal means to facilitate pairwise comparison between groups using a reference grid consisting of combinations of factor levels, with each covariate set to its mean value (*Leonard et al., 2020*). It helps differentiate between the different factors under considerations and identify potential interaction type shifts along the different units of assessment (in this case ecosystems and countries) (*Dangles et al., 2018*).

In order to if and how the relative importance of climatic, topographic, forest structure and human modification variables in driving fire dynamics during the wet and dry seasons of the El Niño and non-El Niño years, a machine learning model, Multivariate
Adaptive Regression Spline (MARS) was used to identify the relative importance of these for the different scenarios—(a) dry season in El Niño years (b) wet season in El Niño years (c) dry season in non-El Niño years (d) wet season in non-El Niño years. MARS is underpinned by the construction of smoothing functions (spline) based non-linear regression models. MARS follows the generic form:

$$y = f(x) + error \tag{1}$$

Here $y$ is the response variable (the different fire variables) and $x$ are the bioclimatic, land use and anthropogenic predictors. The predictors are incorporated into the MARS structure as part of a function (see Eq. (2)) generating a model for the dependent variable. The unknown regression function, $f(x)$ is derived as follows:

$$f(x) = \beta_0 + \sum_{m=1}^{M} \beta_m B_m(x) \tag{2}$$

where $\beta_0$ is the intercept of the model, $\beta_m$ is the coefficient of the $m$-th basis function, $M$ is the number of basis functions in the model and $B_m(x)$ is the $m$-th basis function. Each $B_m(x)$ takes one of the following two forms: (i) a hinge function of the form max $(0, x\_k)$ or max $(0, k\_x)$, where $k$ is a constant value; (ii) product of two or more hinge functions that, therefore, can model the interaction between two or more predictors ($x$) (*López-Serrano et al., 2016*).

MARS creates an initial an overfitted model which includes all the predictor variables. In the next step, a backward procedure is applied to prune the model by removing the predictors one by one, by deleting the least useful predictor at each iteration. The final parsimonious model with the most useful/informative predictors is retained. The contribution of the effective predictors make to the final model is further assessed using the following criteria: (i) nsubsets, *i.e.*, the number of times that each variable is included in a subset (in the final model); (ii) sqr-rss, *i.e.*, the decrease in the sum of square errors (RSS) for each subset relative to the previous subset; and (iii) sqr-gcv, *i.e.*, the decrease in the GCV value for each subset relative to the previous subset (*López-Serrano et al., 2016*). The MARS model was implemented using the caret package of the R programming language (*Kuhn, 2008*). Residual Sum of Squares (RSS) was used for quantifying the importance of the different predictors in explaining the variation in the response variable(s). The univariate response curves of the most influential predictor variables were derived using partial dependence plots (PDPs), to help visualize the direction of the relationship between the response and the given predictor variable (*Singh et al., 2017*).

## RESULTS

### Variation in fire attributes for different Amazonian countries

While the four factors, month type, season type, landcover and forest structure individually had a significant impact on the different fire attributes, the interaction between these factors did not. The interaction between the month and season type was statistically significant for the different fire attributes (Fig. 2) for the Amazon basin.
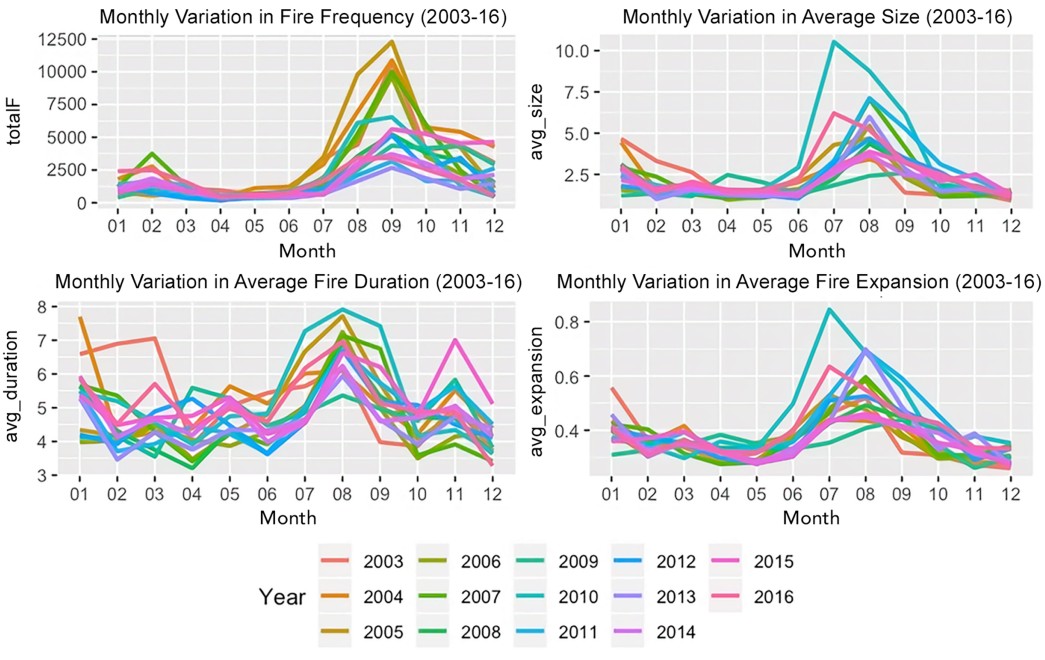

**Figure 2 Monthly variation in fire characteristics from 2003–2016.**

Post-hoc analysis revealed that the main differences existed between the fire dynamics of croplands *vs.* the others for fire size and expansion. Fire expansion was also different between evergreen broadleaf forests and savannas. The role of year type, month type and forest structure and their interactions varied across the landcovers of different countries. The fire expansion in the grasslands of Colombia, the fire size and duration in the croplands of Brazil, fire size and duration in the savannas and grasslands of Bolivia were influenced by the year type, *i.e.*, whether it was an El Niño year or not. Further the fire size and duration in the croplands of Peru and the fire duration in the evergreen broadleaf forests of Guyana were also influenced by the year type. Forest structure was a significant factor contributing to fire size across all the landcover classes (except croplands) in Brazil and for all the landcover classes in Colombia and Bolivia. Except for grasslands and woody savannas, the season and year type interactions were significant factors in influencing fire duration in Brazil while this interaction was only significant for the grasslands of Bolivia. The season type was a significant factor for influencing fire size across all landcovers except the evergreen broadleaf forests of Guyana.

## Bioclimatic driver variations

Linear mixed effects modelling revealed that for each of the bioclimatic variables under consideration that while year type was not significant while the season type (wet season *vs.* dry season) was significantly important (see Fig. 3). The soil moisture of the woody savannas of Colombia and the drought severity in the croplands of Peru were influenced by the year type, *i.e.*, whether it was an El Niño year or not. For savannas and grasslands of Colombia, the interaction between the year type and month type were significant in

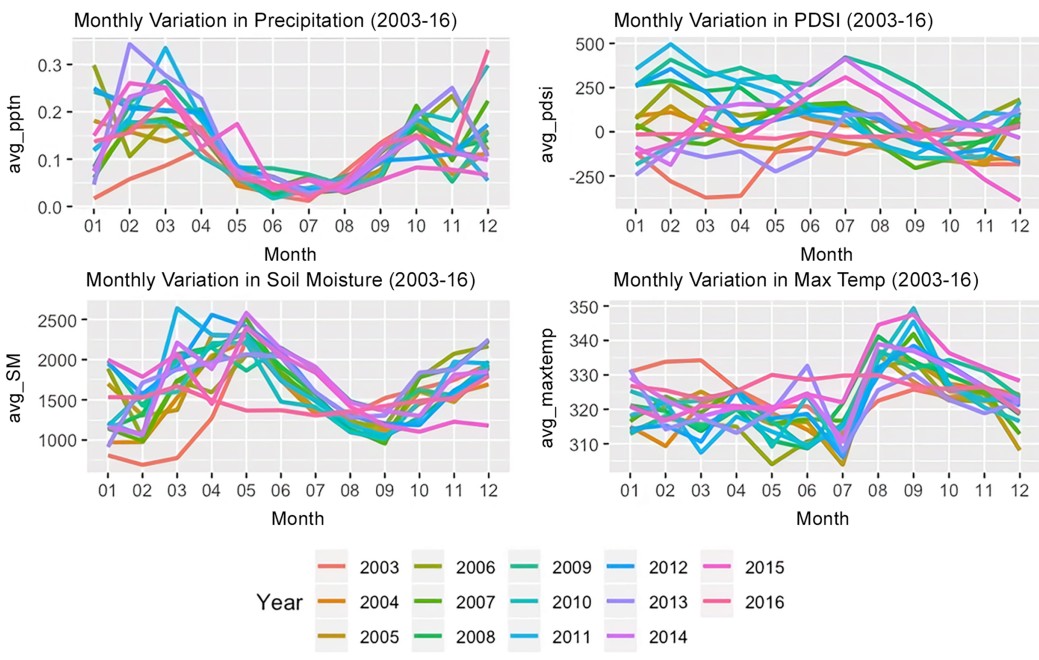

**Figure 3 Monthly variation in bioclimatic characteristics from 2003–2016.**

influencing both the soil moisture, drought severity and maximum temperature. In Brazil, the soil moisture was significantly influenced by the forest structure, month type and the interaction between the year and month type for all the landcover types while all these factors were significant only for evergreen broadleaf forests and croplands in the case of precipitation.

The interactions between the year type, season type, land cover and forest strata/ structure were significant for all the bioclimatic variables under consideration when the entire study area was considered. Post-hoc analysis revealed that the main differences existed between the bioclimatic variables of croplands *vs*. the others, savannas *vs*. evergreen broadleaf and woody savannas, and grasslands *vs*. evergreen woodlands. An examination of how the role of the year type, month type and forest structure varied across the landcovers of different countries revealed that the forest structure was a significant factor in influencing the variation in bioclimatic variables for most of the situations. The interaction between the season type and year type was significant for maximum temperature in the case of all landcover types in Brazil and only for woody savannas in Peru. The interaction between the year type and season type was not a significant driver in the case of the different landcover types of Peru and Guyana for soil moisture. This interaction was significant only for precipitation over woody savannas and the maximum temperature and drought severity of evergreen broadleaf forests of Peru and the croplands of Guyana. Soil moisture for all the landcover classes of Bolivia were significantly influenced by forest structure, season type and the interaction between season and year type.

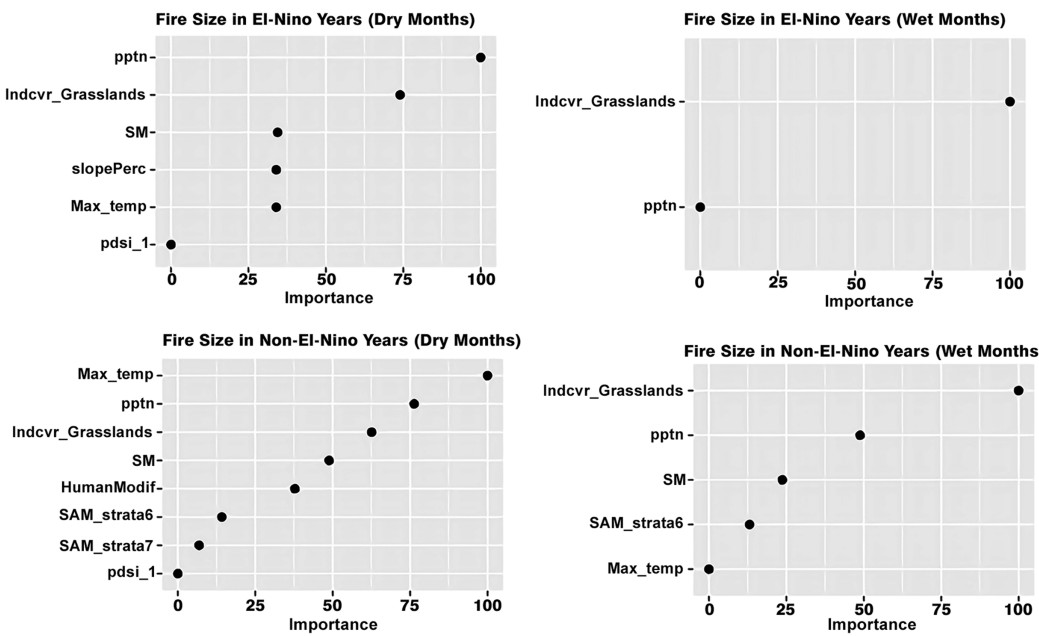

**Figure 4  Variables influencing fire size across the different scenarios.**

## Role of bioclimatic, land cover and anthropogenic variables in influencing fire attributes

Machine learning modeling revealed that relative importance of bioclimatic, land cover and human modification in influencing the different fire dynamics varied across all four scenarios—(a) dry season in El Niño years (b) wet season in El Niño years (c) dry season in non-El Niño years (d) wet season in non-El Niño years. While precipitation and presence of grasslands are the most important determinants of fire size in the dry months of the El Niño years, grasslands alone were the most important determinant of fire size (Fig. 4).

Grasslands are the most important drivers of fire speed in both the wet and dry seasons of El Niño years as well, along with the wet non-El Niño months. While maximum temperature, precipitation and soil moisture are important drivers in explaining the variation in fire size in the dry season of the non-El Niño years, the maximum temperature did not explain the fire size variation in the non-El Niño years. The presence of dense and tall forest strata (strata 6) was an important driver for the non-El Niño years.

Slope percentage and maximum temperature were important drivers of fire speed in dry seasons while precipitation and evergreen forests were important drivers of fire speed in the wet season of El Niño years. The maximum temperature was the least important (Fig. 5).

Grasslands were the most important drivers of fire speed for both the dry and wet seasons of the non-El Niño years, along with soil moisture and maximum temperature. In addition to grasslands, maximum temperature, the presence of evergreen broadleaf forests, human modification and precipitation too were important drivers of fire speed in

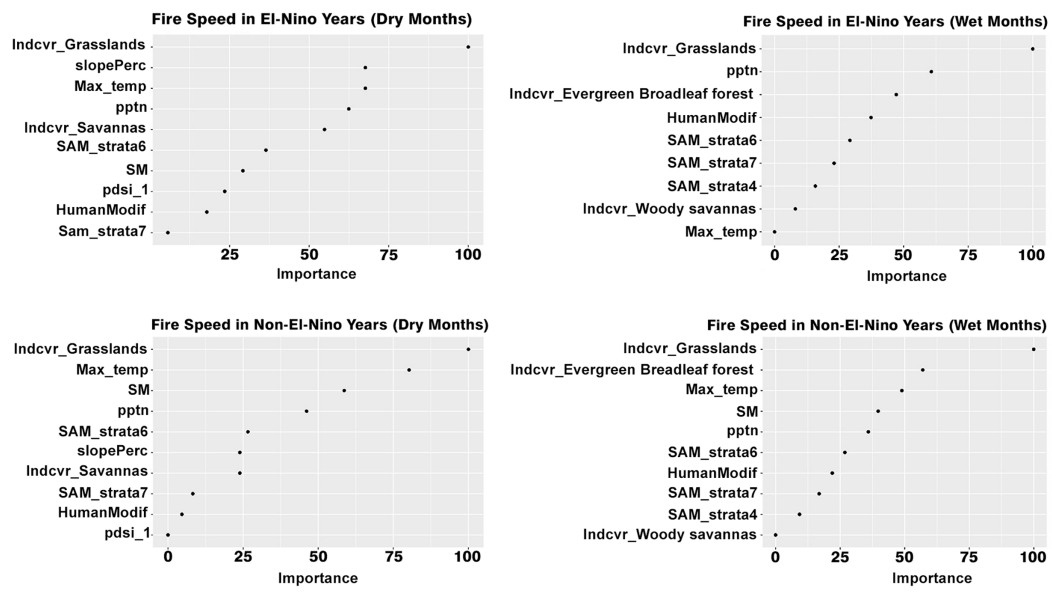

**Figure 5** Variables influencing fire speed across the different scenarios.

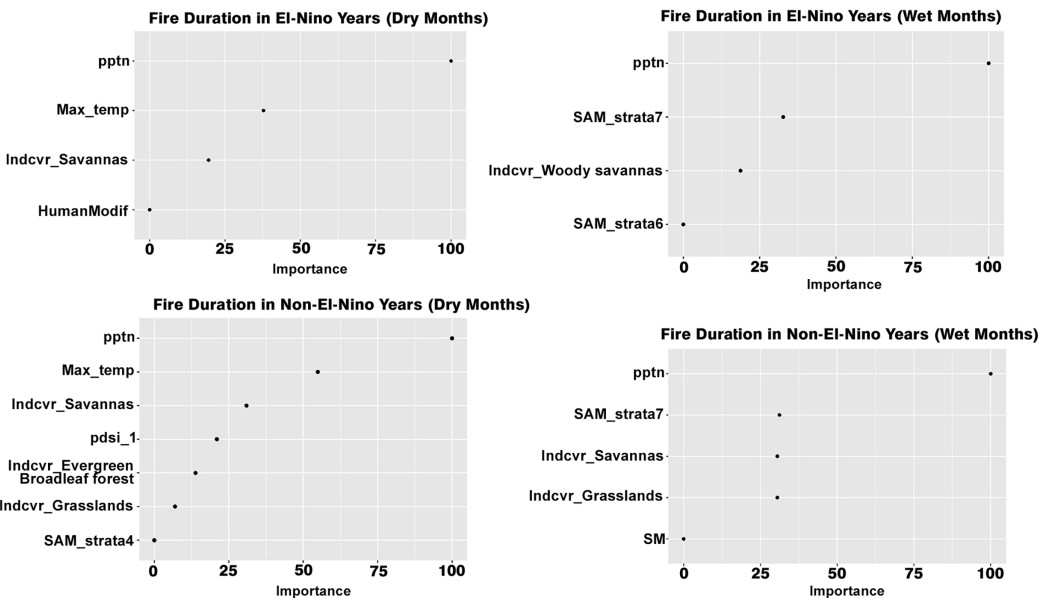

**Figure 6** Variables influencing fire duration across the different scenarios.

the wet months of both El Niño and non-El Niño years. Precipitation was the most important driver of fire duration for all four scenarios. The maximum temperature was the most important driver of fire duration in the dry seasons of both the El Niño and non-El Niño years (Fig. 6).

The presence of savannas was an important determinant of fire duration for all cases. Presence of grasslands was an important determinant of fire expansion in all the scenarios

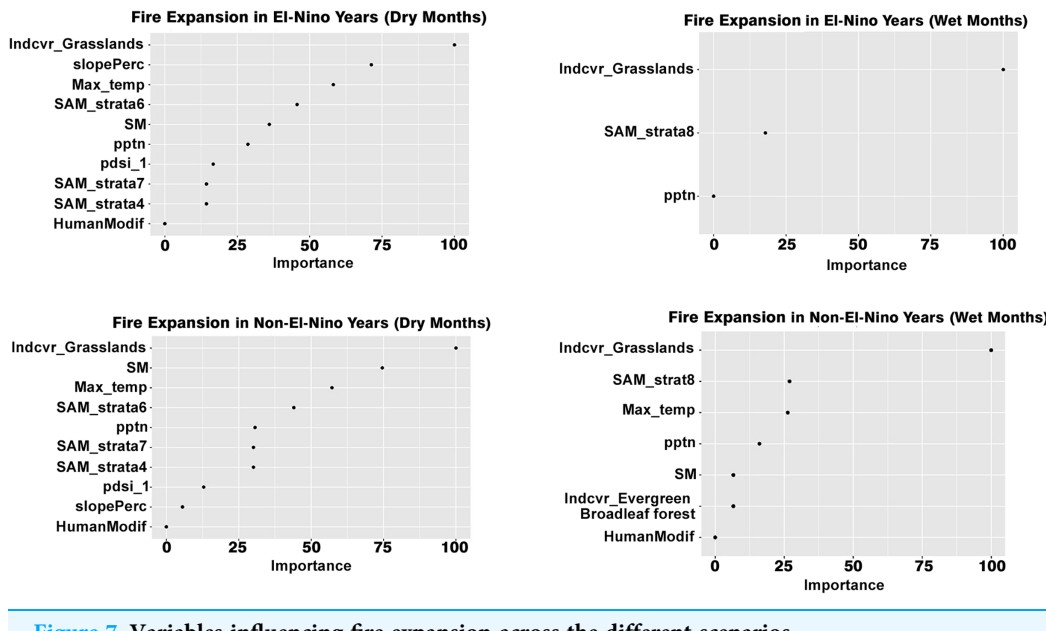

**Figure 7 Variables influencing fire expansion across the different scenarios.**

while the presence of barren earth was an important determinant for the wet season of both the El Niño and non-El Niño years (Fig. 7).

Maximum temperature and soil moisture were the most important determinants of fire expansion in the dry seasons of both El Niño and non-El Niño years. The human modification did not impact forest fire expansion in any of the scenarios. The average fire expansion for the grassland land cover was 0.88 km/day in the dry months of the El Niño years and 0.4 km/day for all other ecosystems. For the wet months of the El Niño years, average fire expansion for the grassland land cover was 0.56 and 0.3 km/day for the non-grassland ecosystems. For the non-El Niño years, the average fire expansion was 0.9 km/day for grasslands and 0.42 km/day for non-grasslands in the dry season and 0.52 and 0.3 km/day respectively for the dry season.

Partial dependence plots revealed that for all the scenarios, precipitation and soil moisture were inversely related to fire size and speed while the maximum temperature was proportionally related to fire size, meaning that lower precipitation, lower soil moisture and higher temperatures contributed to higher fire sizes. Precipitation was inversely related to fire duration in the El Niño and non-El Niño years (both dry and wet months). Soil moisture too was inversely related to fire dynamics, for both El Niño and Non-El Niño years. The high human modification index (>0.25) had a positive association with fire speed in the wet seasons of the El Niño years. Dense forest structures (especially strata 6 and 7) had a negative association with fire dynamics such as fire speed and duration. See Supplementary Materials for more details.

## DISCUSSION

While the role of precipitation in increasing fire frequency in western Amazonia has been identified in previous research, our research findings indicate that precipitation can

influence fire dynamics such as size, duration and speed across the Amazonian biome. For instance, fire size and duration were inversely related to precipitation for both the dry and wet months in the El Niño and non-El Niño years, indicating that lower precipitation resulted in larger fires that burnt for a longer time. Our findings are congruent with the existing body of literature. El Niño events are associated with decreased precipitation and increased temperature and drought severity (*Moura et al., 2019*; *Da Silva Júnior et al., 2019*) which has a cascading impact on fire dynamics (*Mota et al., 2019*). Landcover and topographic variables too are important contributors to fire risk; with pasture and cropland dominated areas seeing increased fire risk in parts of the Brazilian Amazon. Our research has identified the role of both landcover and bioclimatic variables in influencing fire dynamics, including quantifying how the interactions between different drivers varies across the different Amazonian countries and identifying the relative importance of these in influencing fire dynamics. We have taken a country specific approach as opposed to a regional approach taken by other researchers (*Mota et al., 2019*; *Da Silva Júnior et al., 2019*) to support the development of fire management strategies at an inter and intra-country scale in the Amazonian biome.

## Role of landcover type in influencing fire dynamics

For both bioclimatic and fire variables, the significance of the year type, seasonality, land cover and forest types and their interactions varied across the Amazonian countries under consideration. The interaction between climatic factors and fire occurrences varies across the Amazonian basin and is often mediated by anthropogenic factors, such as deforestation (*Armenteras & Retana, 2012*). Our analysis indicates that for both bioclimatic and fire variables, the significance of the year type, seasonality, land cover and forest types and their interactions varied across the Amazonian countries under consideration. Moreover, the presence of grasslands was found to be an important variable in explaining the variation in different fire dynamics across the different Amazonian countries.

The role of natural and anthropogenic variables in influencing fire dynamics varies globally (*Ying et al., 2018*). Factors such as bioclimatic variables (precipitation and drought) (*Aragão et al., 2007*), landcover change dynamics (*Kelley et al., 2021*), seasonality and El Niño impacts are among the most widely used variables used to study Amazonian fire dynamics (*Da Silva Júnior et al., 2019*). Many fire dynamics such as size, speed and expansion were greatly enhanced in grasslands, especially during the dry seasons of the El Niño years. Thus, the presence of grasslands was found to be an important variable in explaining the variation in different fire dynamics. In eastern Amazonia, pasture burning was a significant driver of forest fires, fires that origin in grassy pastures can penetrate inside forests owing to edge conditions (*Cochrane & Laurance, 2002*). These landcover changes have an impact on local microclimatic conditions and the interactions of these can result in elevated fires, especially in degraded forested ecosystems (*Ray, Nepstad & Moutinho, 2005*) and ecosystems dominated by pastures or croplands (*Mota et al., 2019*). Grasslands in the study area mostly consisted of bare earth/sparse

vegetation and low cover forest strata (Supplemental 1) and pasture development is one of the most common land use types that follows deforestation (*Morton et al., 2006*).

In addition to landcover types, we also included forest strata (structure in terms of canopy cover and intactness) as an explanatory variable. Dense and relatively undisturbed forests were found to have the lowest fire activity. More disturbed forested tend to have lower humidity retention and higher temperature (*Silva Junior et al., 2018*) which arguably explains why the precipitation and temperature of disturbed ecosystems such as grasslands/pastures and croplands were influenced by seasonality and had elevated fire dynamics. Changes in landcover structure and management have been found to be significant drivers of fire activity in the Amazonian biome (*Kelley et al., 2021*); for example, changes in forest structure because of forest removal is a major driver of forests fires in the Brazilian Legal Amazon (*Tyukavina et al., 2017*). This research too identified that forest structure influenced fire characteristics; characteristics such as fire duration were lowest in dense forests and highest in grass-dominated ecosystems across the different parts of the Amazonian biome.

The creation of croplands and grass-dominated ecosystems is a major perverse outcome of deforestation in Amazonia (*Shimabukuro et al., 2014*). In both the Brazilian Legal Amazon and the Colombian Amazon, majority of the forest conversions resulted in pasture creation (*Tyukavina et al., 2017*; *Murillo-Sandoval et al., 2018*; *Jakimow et al., 2018*). Grass-dominated vegetation has replaced degraded forests in many parts of the Amazonian basin (*Veldman & Putz, 2011*). For instance, both native and invasive grasses colonized the open canopied forests which had undergone degradation because of forest degradation (*Veldman et al., 2009*). Croplands and pasture-dominated ecosystems were found to be at an elevated fire risk in parts of the Brazilian Amazon (*Mota et al., 2019*). The development of these grass-dominated ecosystems, arguably, has played an important role in both driving and accelerating fire dynamics (in terms of size, speed and expansion). Fires are used for land clearing, agriculture and grassland maintenance and the spillage of fires from adjoining pastures has become an important cause of fires within forested ecosystems of the Amazonian basin (*Alvarado et al., 2004*). Since forest structure is significantly influenced by anthropogenic activities; those variable acts as a proxy for anthropogenic impact.

Previous research has established that grass-dominated ecosystems have the highest rates of burning and more intense fires were observed in grasslands (*Foltz & McPhaden, 2008*). This research identified grass-dominated ecosystems had greater fire sizes, speed and duration, especially during the dry seasons. It may be inferred as more and more of the Amazon rainforest is deforested and converted to agricultural pastures and subjected to other anthropogenic perturbations, human modified landscapes may be more vulnerable to more intense fire events (*Withey et al., 2018*). Dense forests have been discovered to lower rates of burning as compared to open and transitional forests (*Alencar et al., 2015*). This research too identified that fire characteristics such as fire duration were lowest in dense forests and highest in grass-dominated ecosystems across the Amazonian biome. Additionally, the role of bioclimatic variables in driving these

variables across the different seasons and year-types has been quantified, which can arguably contribute to extra vigilance during those phases.

## Bioclimatic drivers and their role in influencing fire dynamics

Bioclimatic variables such as precipitation are a major driver of forest fires across all the biomes of South America (*Balch et al., 2015*), including the Amazon basin. The role of bioclimatic variables in influencing Amazonian fire frequency has been established in the existing body of research (*Xu et al., 2020*). Our research also establishes that precipitation can influence other fire dynamics such as size, duration and speed. We also discovered that precipitation declined notably in the dry season for all the years and notable precipitation declines observed in August and September in different parts of the Brazilian Amazon (*Silva Junior et al., 2019*) and precipitation undergoes a sharp decline during the El Niño years (*Moura et al., 2019*). Lack of rainfall during the dry season has a strong influence on increasing fire risk in the region (*Marengo et al., 2018*) as do increased droughts (*Aragao et al., 2007*). We also established that fire size and duration were inversely related to precipitation for both the dry and wet months in the El Niño and non-El Niño years, indicating that lower precipitation resulted in larger fires that burnt for a longer time. Precipitation was previously discovered to have an inverse relationship with different fire attributes in South America (*McWethy et al., 2018*) with lower rainfall resulting in higher incidences of fire. Previous research discovered that decline is precipitation and elevated fire activity from July–September is known to characterize the Brazilian Amazon (*Fonseca et al., 2019*). It may be inferred that the phenomenon of precipitation declines and elevated fire activity from July to September observed in the Brazilian Amazon (*Da Silva Júnior et al., 2019*; *Fonseca et al., 2019*) potentially exists in other Amazonian countries as well. Moreover, drought severity makes grass-dominated and savanna-forest transition zones more vulnerable to fires (*Xu et al., 2020*).

The El Niño phenomenon has a strong impact on the climatic patterns of the Amazon basin (*Malhi & Wright, 2004*). Severe droughts in the area are associated with increased fire activity. The El Niño phenomenon is linked with increased temperatures and drought severity and reduced precipitation (*Moura et al., 2019*), which in turn increases fire frequency (*Silva et al., 2021*). Central Amazonia's forests which are typically regraded as being fire-resistant saw increased fire activity and large fires in the El Niño years (*Da Silva et al., 2018*). Owing to increasing extreme temperatures, drought duration and intensity has worsened in Amazonia, increasing the fire risk in grass dominated savanna ecosystems (*Xu et al., 2020*) and thee remain at a greater risk of elevated fire activity in the El Niño years (*Da Silva Júnior et al., 2019*). Thus, open canopied ecosystems are more vulnerable to drought driven fire frequency increased, especially during El Niño years (*Fonseca et al., 2017*).

This study also establishes that bioclimatic and fire variables including characteristics such as x speed and size were not only influenced by whether it was an El Niño year or not but also by the interaction between the year types, season type and land use related variables. While El Niño events play an important role in influencing bioclimatic and fire dynamics, research indicates that changes in these dynamics may not always be directly

linked to El Niño (*Da Silva Júnior et al., 2019*). Previous research also established that bioclimatic variables varied across the different regions of the Amazon basin; for example, whilst statistically significant negative rainfall trend persists in southern Amazonia during the pre-rainy season, the precipitation situation is opposite in northwestern Amazonia (*Marengo et al., 2018*). Differences in bioclimatic variables, including rainfall, have consequently shown diversity across the different countries of the Amazon basin (*Espinoza Villar et al., 2009*) which influences in variation in fire dynamics (in conjunction with other factors). Arguably, this calls for developing country specific fire mitigation and conservation strategies which can also account for changes in seasonality, presence of El Niño events and ecosystem structural attributes.

## Conservation implications of the research findings

The findings of our research have notable conservation implications. Our research establishes the vulnerability of grass-dominated ecosystems and disturbed strata to fire dynamics. These findings are corroborated by existing literature (*Silva Junior et al., 2018*; *Da Silva et al., 2018*; *Xu et al., 2020*; *Pontes-Lopes et al., 2021*) and fires escaping from pasture dominated areas to forested ecosystems is matter of significant conservation concern (*Cano-Crespo et al., 2015*).

Our research identifies the fire behaviour of different ecosystems across the different Amazonian countries and discovered that dense forests had lower fire incidences, and this is congruent with the existing body of literature; open ecosystems are more vulnerable to fires than dense forests (*Alencar et al., 2015*; *Alencar, Nepstad & Diaz, 2006*). Significant variation in fire dynamics exist in grass-dominated and anthropogenic ecosystems such as croplands, especially during the dry season (*Cano-Crespo et al., 2015*; *Da Silva Júnior et al., 2019*). While what constitutes as a dry season arguably differs across the different parts of the Amazon biome, the months from July–September see elevated fire activity and reduced precipitation and elevated temperatures for most of the Amazonian biome (*Silveira et al., 2020*; *Aragão et al., 2007*; *Yokelson et al., 2007*; *Reddington et al., 2015*; *Heyer et al., 2018*; *Fonseca et al., 2019*; *Da Silva Júnior et al., 2019*). Dry season evapotranspiration from June to September was found to be significantly lower for anthropogenic ecosystems such as pastures as compared to primary forests in the Brazilian Amazon, which arguably contributes to their fire risk (*Khand et al., 2017*). Dense forests showed a higher level of resilience to fire events (*De Andrade et al., 2020*). Hence, on this basis it can be suggested that grass-dominated ecosystems and croplands be monitored closely during these months, especially during the El Niño years to prevent leakage into forested ecosystems. Since the phenomenon observed in the Brazilian Legal Amazon has been demonstrated in other anthropogenic ecosystems of the different Amazonian countries we included for our analysis, it makes sense that the conservation measures include these areas as well. Expansion of pasturelands in the Brazilian Legal Amazon has contributed to an increase in fire activity per square km (*Libonati et al., 2021*). Arguably, the phenomenon of anthropogenic ecosystems contributing such as pastures contributing to increased fire activity is playing out in other Amazonian countries. Detection of understory fires is a significant challenge encountered in the field of Amazonian fire

studies; these are difficult to detect owing to their smaller sizes and lower intensities (*Asner & Alencar, 2010*; *Da Silva Júnior et al., 2019*). However, research suggests that these too get accelerated during the drought season (*Balch et al., 2011*). Arguably drought seasons, especially intense El Niño drought seasons pose a significant risk to the different Amazonian ecosystems and increased interventions during this period should be considered.

Furthermore, elevated anthropogenic disturbances result in elevated fire dynamics. This finding has been borne by a body of literature (*De Andrade et al., 2020*; *Dos Reis et al., 2021*). Arguably, composite datasets such as the global human modification index (which is based on a cumulative measure of human influence based on eight global human pressures), including road networks (*Venter et al., 2016*) too can be used for identifying areas at an elevated risk of fires across the Amazonian biome and inform conservation pathways as done by *Singh & Yan (2021)*.

## Caveats of the research

Amazonian fire dynamics are influenced by a multitude of bioclimatic and landcover related factors and complex inter-linkages between these with the nature and depth of these interlinkages not completely understood. While our study and many other studies have considered El Niño as a static contribution factor (*Fonseca et al., 2017*; *Xu et al., 2020*; *Armenteras et al., 2017*), El Niño is a dynamic phenomenon (*Espinoza et al., 2014*; *Jimenez et al., 2019*). While El Niño induced droughts have exacerbated fire activity in the Amazonian biome but each of these have a distinct pattern of their own influenced by a configuration of climatic variables (*Silva et al., 2021*). The El Niño phenomenon is influenced by a variety of other factors, for instance land surface and ocean temperatures which in turn can influence fire dynamics. For instance, the extreme droughts of 2015 were perpetuated as result of the abnormal heating of the Tropical Atlantic Ocean which increased drought severity and higher fire frequency (*Da Silva Júnior et al., 2019*). Arguably, considering the nature and intensity of the different El Niño events in fire studies can help deepen our understanding of the complex inter-linkages that underpin Amazonian fire dynamics. In addition, the impacts of El Niño are compounded by anthropogenic effects (*Bush et al., 2017*) and El Niño also impacts the dry season (*Armenteras et al., 2017*). Dry season has been identified as playing an important role in influencing fire dynamics even though the exact length of dry season is influenced by a variety of factors and remains variable (*Balch et al., 2015*). Our research identifies role of other factors, including seasonality and its interactions with El Niño in influencing fire dynamics across the different Amazonian ecosystems.

While we have considered a subset of variables to explain the variation in fire dynamics, many other variables, related to the variables considered in our study influence fire dynamics. For instance, hydrological cycle too plays an important role in influencing fire dynamics (*Xu et al., 2020*). Hydrological cycle in turn is influenced by droughts and extreme climatic phenomenon such as El Niño (*Da Silva Júnior et al., 2019*). Droughts also result in declining soil moisture and ensuing tree mortality increased the fuel load which contribute to elevated burning (*Berlinck & Batista, 2020*). Given the overwhelming

importance of drought and precipitation patterns in influencing fire behavior (*Asner & Alencar, 2010*), these were the variables included in our research. Including all those interactions is beyond the scope of our study. hence it is important to interpret our findings and those of other researcher in the context of these caveats to improve result interpretability and derive conservation centric insights.

## CONCLUSION

While most studies on the Amazon basin focused on the Brazilian Amazon, this study established the variability of bioclimatic variables and fire dynamics in the different countries that have the Amazon rainforest within their boundaries, *i.e.*, Brazil, Bolivia, Peru, Colombia, Guyana and Ecuador. Fire characteristics (fire size, fire speed, fire duration and expansion) and their bioclimatic drivers (precipitation, maximum temperature, soil moisture and drought severity) varied between dry and wet seasons of the Amazon biome from 2003–2016. Declining precipitation and increased temperatures have a strong impact on the fire characteristics for El Niño years, which also saw greater fire sizes and speeds as compared to non-El Niño years.

The presence of grass-dominated ecosystems was an important driver of the different fire dynamics and these ecosystems recorded higher fire sizes compared to dense canopied forests. These findings can help develop appropriate ecosystem specific interventions and management practices to control fires, especially in the El Niño years. We have only considered a subset of the most used drivers in our research. These by no means are not the only drivers that influence the fire dynamics in the Amazonian biome. Fire dynamics in the Amazonian biome is a deeply complex phenomenon influenced by a variety of other considerations such as the role of ocean currents in influencing El Niño to socioeconomic factors influencing landcover change. Covering all of these is beyond the scope of our research. However, the frameworks and methodology workflows developed in our research can facilitate the inclusion of other parameters in future studies and improve fire monitoring capabilities in the region.

### Funding
This research was conducted as a part of the Imperial-TUM seed fund. The funders had no role in study design, data collection and analysis, decision to publish, or preparation of the manuscript.

### Grant Disclosures
The following grant information was disclosed by the authors:
Imperial-TUM seed fund.

### Competing Interests
The authors declare that they have no competing interests.
## Author Contributions

- Minerva Singh conceived and designed the experiments, performed the experiments, analyzed the data, prepared figures and/or tables, authored or reviewed drafts of the paper, and approved the final draft.
- Xiaoxiang Zhu conceived and designed the experiments, authored or reviewed drafts of the paper, and approved the final draft.

## Data Availability

The raw data are available in the Supplemental File.

## Supplemental Information

Supplemental information for this article can be found online at http://dx.doi.org/10.7717/peerj.12029#supplemental-information.

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
