# Peer review of "Analysis of how the spatial and temporal patterns of fire and their bioclimatic and anthropogenic drivers vary across the Amazon rainforest in El Niño and non-El Niño years"

_PeerJ, doi:10.7717/peerj.12029_

## Round 0.1 · original submission · Major Revisions

All three anonymous reviewers have suggested major revisions to your manuscript. They have identified serious issues that need to be addressed. The manuscript requires a thorough re-write and re-analysis of the data. Some questions have been raised about the validity of the conclusions derived from your study. Please pay particular attention to the remarks from all three reviewers and in your next revision please specify the changes that you have made and how you responded to each of the questions raised. This manuscript will require re-review.

Reviewer 1 ·

Basic reporting

This manuscript is well written and used professional English. The most prominent literature has been cited in this paper.

Experimental design

The experiment is well designed. However, surface fires under the tree canopy are difficult to be detected through MODIS, which means the authors could have missed the surface fires. It deserves a discussion on how the missed surface fire could affect your results. Additionally, the Amazon region is experiencing frequent clouds, which could degrade remote sensing quality. How did you solve this issue?

Validity of the findings

Data quality needs a thorough discussion.

·

Basic reporting

This research topic is relevant and interesting. The Abstract could provide more information to the reader of why this particular study is important and needed. More context and careful interpretation of the literature are needed to improve both the Abstract and Introduction. Also, many variables and metrics were presented in the Abstract and Introduction setting this study to be a very ambitious one, however, without cohesion and focus it became very confusing to understand properly what was being assessed and how all the variables were connected. It was challenging to assess if the chosen data and statistical analysis used were appropriate for this study because the objectives of this investigation were not clear at all. The results seemed interesting and instigated that this study has potential, but it was impossible to assess if the results had hit the mark once I did not know what we were aiming at. I strongly encourage the authors to work on this manuscript focusing on improving the clarity and focus of the study to simplify the message using only what is needed to deliver it, and thus make it a stronger paper.

Experimental design

The research question seems to be relevant and meaningful, however it is not well defined and explained in this manuscript. The Methods can be improved but more importantly they need to be better connected with the Introduction. It was difficult to evaluate if the Data and Methods are appropriate without knowing clearly what was being investigated.

Validity of the findings

Currently it is impossible to review and assess if the findings and the discussion around them are relevant or well stated as I cannot see how they are well linked with the original research question.

Additional comments

This is a promising study, and it seems the authors have good knowledge of statistics to provide solid analysis. Unfortunately, it was impossible to dive in the analysis and truly assess the results and interpret the discussion as I could not understand what was being investigated. I strongly encourage the authors to improve the packaging of this study and try and make the message very clear. I will be happy to review this manuscript in the future once the whole study is more cohesive.

·

Basic reporting

i) The English seem adequate.
ii) I identify a lack of understanding of the drivers of Amazonian droughts. There are a vast literature about it, one chan check José Marengo papers.
iii) The study focus on an important issue and has the potential to contribute to the scientific discussion, but there are two major problems in their design, which is described in the next section.

Experimental design

The article is interesting but there are three major problems in the experiment design, described below.
ii) I understand that there is an insufficient literature background, which deeply affects the methods: The authors define El Niño x non-El Niño years. Although El Níno is an important drought driver in Amazonia, this is not the single driver acting during the time-series studied. Large-scale weather patterns affect the incidence of rain in the Amazon. The essential driver of the hydrological cycle is water from the oceans. When heated, in intra- and inter-annual and decennial cycles in an anomalous manner, they cause changes in the large-scale atmospheric circulation and may intensify or attenuate the rain patterns on the continent. A clear connection is observed in the relationship between the surface temperature of the oceans and the rain: When the Atlantic Ocean is warmer than normal climatological conditions, in the region of the Atlantic Multi-Decadal Oscillation (AMO index), there is a decrease in the rain in the southwestern quadrant of the Amazon; When the Pacific is hotter than normal, in the El Niño phenomenon (MEI index), the reduction of rain occurs in t the central, eastern and southern parts of the Amazon. Finally, when the Ocean in the Pacific Decadal Oscillation region (PDO index) is hotter than normal, rainfall reduction is concentrated in the central part of the Amazon. Not accounting for these interactions properly, affects the climatological analysis, which becomes an incorrect representation of the spatial and temporal processes associated with droughts at the paper results.
iii) The dry and wet season definition from August to October is simply not adequate - the dry season in the northern hemisphere starts in November and goes up to March, as well as the fire peaks. Therefore, the analysis is incorrect for the North part of the biome above the equator (Venezuela, part of Colombia, Guyana..). Even in Southern Amazon, the dry season varies.
iv) Depending on when, during the year, the drought occurred (drought during the dry season or drought during the wet season) will deeply affect the fire characteristics and the forest vulnerability to the fire. Thus, the fire "accountability" in the paper results may be incorrectly captured.

Validity of the findings

The problems identified in the section above mean that the results are compromised, and must be completely revisited. Experiment design must be changed.

Additional comments

This paper aims to bring an important contribution with a method that has some potential. However, the problems in the experiment design make the results unreliable.

In addition to accounting for the dry years process correctly, which currently it is not, you also need to take into account the dry season and wet season correctly; and also, it must be identified when the drought hit the region/forest. For example, if the drought occurs during the dry season, it may slightly exacerbate the forest vulnerability to fires, but if the drought hits during the wet season, it means that the forest will enter the dry season already in water deficit, exacerbating its fire vulnerability. This is clearly what happened in southwestern Amazonia during the 2015/16 El Nino. Perpahs, instead of rainfall, you can use Water Deficit to better capture this and thus disentangle the drivers, as you tried to perform. By looking at rainfall only, the western part of the Amazon practically has no dry season, but one needs to look at the water deficit, which is better related to the forest vulnerability to fires due to water stress.

Some must-read papers:

BERENGUER E, CARVALHO N, ANDERSON Liana O., ARAGÃO LEOC, FRANÇA F, BARLOW J. Improving the spatial‐temporal analysis of Amazonian fires. Global Change Biology, letter to the editor , https://doi.org/10.1111/gcb.15279.

PESSÔA, A.C.M.; ANDERSON, Liana O.; CARVALHO, N.S.; CAMPANHARO, W.A.; JUNIOR, C.H.L.S.; ROSAN, T.M.; REIS, J.B.C.; PEREIRA, F.R.S.; ASSIS, M.; JACON, A.D.; OMETTO, J.P.; SHIMABUKURO, Y.E.; SILVA, C.V.J.; PONTES-LOPES, A.; MORELLO, T.F.; ARAGÃO, L.E.O.C. Intercomparison of Burned Area Products and Its Implication for Carbon Emission Estimations in the Amazon. Remote Sens. 2020, 12, 3864. https://doi.org/10.3390/rs12233864

ANDERSON, L.O.; RIBEIRO NETO, G.; CUNHA, A.P.; FONSECA, M.G.; MENDES, Y. M. ; DALAGNOL, R.; WAGNER, F.H.; ARAGÃO, L.E.O.C. Vulnerability of Amazonian forests to repeated droughts. Philosophical Transactions of the Royal Society B: Biological Sciences, 2018 373 20170411; doi:10.1098/rstb.2017.0411, 2018.

ARAGÃO, L.E.O.C.; ANDERSON, L.O.; FONSECA, M.G.; ROSAN, T.M.; VEDOVATO, L.; WAGNER, F.; SILVA, C.; JUNIOR, C.; ARAI, E.; AGUIAR. A.P.; BARLOW, J.; BERENGUER, E.; DEETER., M.; DOMINGUES, L.; GATTI, L.; GLOOR, M.; MALHI, Y.; MARENGO, J.; MILLER, J.; PHILLIPS, O.; SAATCHI, S. 21st Century drought-related fires counteract the decline of Amazon deforestation carbon emissions. Nature Communications, 9, ( 536), doi:10.1038/s41467-017-02771-y

ARAGÃO, L. E. O. C.; MALHI, Y.; ROMAN-CUESTA, R. M.; SAATCHI, S.; ANDERSON, LIANA O.; SHIMABUKURO, Y. E. Spatial patterns and fire response of recent Amazonian droughts. Geophysical Research Letters, v. 34, p. doi:10.1029/200, 2007.

SILVA JUNIOR CHL; ARAGÃO L; ANDERSON Liana O.; FONSECA M; SHIMABUKURO Y; VANCUTSEM C; ACHARD F; BLEUCHE R; NUMATA I; SILVA C; MAEDA E; LONGO M; SAATCHI S. Persistent collapse of biomass in Amazonian forest edges following deforestation leads to unaccounted carbon losses. Science Advances 30 Sep 2020, Vol. 6, no. 40, eaaz8360, DOI: 10.1126/sciadv.aaz8360

SILVEIRA, M.V.F.; PETRI, C.A.; BROGGIO, I.S.; CHAGAS, G.O.; MACUL, M.S.; LEITE, C.C.S.S.; FERRARI, E.M.M.; AMIM, C.G.V.; FREITAS, A.L.R.; MOTTA, A.Z.V.; CARVALHO, L.M.E.; SILVA JUNIOR, C.H.L.; ANDERSON, Liana O.; ARAGÃO, L.E.O.C. Drivers of Fire Anomalies in the Brazilian Amazon: Lessons Learned from the 2019 Fire Crisis. Land 2020, 9, 516. https://doi.org/10.3390/land9120516

ARAGÃO, L.E.O.C.; ANDERSON, L.O.; FONSECA, M.G.; ROSAN, T.M.; VEDOVATO, L.; WAGNER, F.; SILVA, C.; JUNIOR, C.; ARAI, E.; AGUIAR. A.P.; BARLOW, J.; BERENGUER, E.; DEETER., M.; DOMINGUES, L.; GATTI, L.; GLOOR, M.; MALHI, Y.; MARENGO, J.; MILLER, J.; PHILLIPS, O.; SAATCHI, S. 21st Century drought-related fires counteract the decline of Amazon deforestation carbon emissions. Nature Communications, 9, ( 536), doi:10.1038/s41467-017-02771-y

SILVA JUNIOR, C. H. L.; ARAGÃO, L.E.O.C.; FONSECA, M.G.; ALMEIDA, C. T.; VEDOVATO, L. B.; ANDERSON, L.O. Deforestation-Induced Fragmentation Increases Forest Fire Occurrence in Central Brazilian Amazonia. Forests 2018, 9(6), 305; https://doi.org/10.3390/f9060305.

The role of ENSO flavours and TNA on recent droughts over Amazon forests and the Northeast Brazil region JC Jimenez, JA Marengo, LM Alves, JC Sulca, K Takahashi, S Ferrett, ...
International Journal of Climatology

Extreme seasonal climate variations in the Amazon basin: droughts and floods
JA Marengo, ER Williams, LM Alves, WR Soares, DA Rodriguez
Interactions between biosphere, atmosphere and human land use in the Amazon …

The extreme 2014 flood in south-western Amazon basin: the role of tropical-subtropical South Atlantic SST gradient JC Espinoza, JA Marengo, J Ronchail, JM Carpio, LN Flores, JL Guyot
Environmental Research Letters 9 (12), 124007

Drought in Bolivia: The worst in the last 25 years JA Marengo, JC Espinoza, LM Alves, J Ronchail
https://doi. org/10.1175/2018BAMSStateoftheClimate. 1

Extreme seasonal droughts and floods in Amazonia: causes, trends and impacts
JA Marengo, JC Espinoza
International Journal of Climatology 36 (3), 1033-1050

---

## Round 0.2 · accepted · Accept

Your article is herewith accepted for publication.